# Learning More by Seeing Less: Structure-First Learning for Efficient, Transferable, and Human-Aligned Vision

## Abstract

Despite remarkable progress in computer vision, modern recognition systems remain fundamentally limited by their dependence on rich, redundant visual inputs. In contrast, humans can effortlessly understand sparse, minimal representations like line drawings, suggesting that structure, rather than appearance, underlies efficient visual understanding. In this work, we propose a novel structure-first learning paradigm that uses line drawings as an initial training modality to induce more compact and generalizable visual representations. We demonstrate that models trained with this approach develop a stronger shape bias, more focused attention, and greater data efficiency across classification, detection, and segmentation tasks. Notably, these models also exhibit lower intrinsic dimensionality, requiring significantly fewer principal components to capture representational variance, which mirrors observations of low-dimensional, efficient representations in the human brain. Beyond performance improvements, structure-first learning produces more compressible representations, enabling better distillation into lightweight student models. Students distilled from teachers trained on line drawings consistently outperform those trained from color-supervised teachers, highlighting the benefits of structurally compact knowledge. Together, our results support the view that structure-first visual learning fosters efficiency, generalization, and human-aligned inductive biases, offering a simple yet powerful strategy for building more robust and adaptable vision systems.

## 1 Introduction

Despite remarkable progress in computer vision, modern recognition systems remain fundamentally constrained by their reliance on dense, high-fidelity visual inputs. State-of-the-art models, particularly those based on deep convolutional and transformer architectures, have demonstrated impressive accuracy across a variety of benchmarks. However, their success is heavily contingent on exposure to vast datasets filled with richly textured, color photographic imagery. This dependency introduces significant inefficiencies—both in terms of data requirements and generalization capabilities—limiting the robustness of these systems when confronted with abstract or atypical visual inputs (Geirhos et al., 2019; Schlegel et al., 2015).

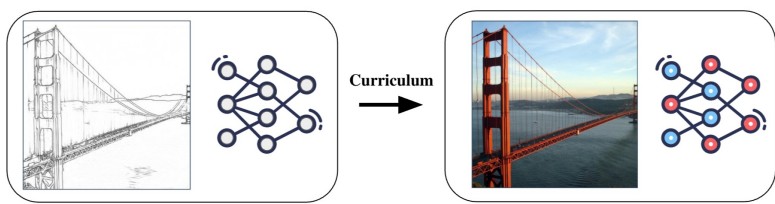

Figure 1: A structure-first curriculum, beginning with line drawings, leads to efficient, transferable, and human-like visual representations.

In contrast, humans can effortlessly interpret line drawings—minimal, structure-preserving representations that strip away redundant surface features. This ability suggests that the human visual

system is tuned to extract and reason over efficient structural information such as shape and topology, rather than relying on unstable texture or color. Neuroscientific fMRI studies support this view, showing that line drawings elicit robust, category-specific activity patterns throughout the visual cortex—comparable to those induced by full-color photographs (Walther et al., 2011). Similarly, Shin et al. (2008) demonstrate that abstract visual symbols like icons also elicits similar neuronal activation in the brain as their corresponding natural images.

Psychological evidence further supports the transformative power of minimal representations in visual learning. Studies reveal that drawing practice enhances perceptual skills and object recognition in humans (Kozbelt, 2001; Chamberlain et al., 2019; Perdreau & Cavanagh, 2015). Artists develop specialized visual cognition through prolonged exposure to structural information—improved visual memory, enhanced structural analysis, and refined attention to diagnostic features (Schlegel et al., 2015; Perdreau & Cavanagh, 2015). These enhancements are believed to be associated with learning from minimal representations (Perdreau & Cavanagh, 2014)—by eliminating distracting surface features, it the visual system is encouraged to extract more robust structural relationships.

Motivated by these insights, we introduce a structure-first learning paradigm. This approach uses line drawings as an initial training modality to guide models toward better representation learning. Our central hypothesis is that by training vision models to first understand the world through minimal structural cues, we aim to induce representations that are more compact, efficient, generalizable, and human-aligned. Thanks to the recent development from Chan et al. (2022), we can now accurately convert photographic color images to their minimal line-drawing form automatically at scale. Our results show that models developed through this structure-first approach exhibit stronger shape bias, more focused attention, and lower intrinsic dimensionality—mirroring characteristic findings of human minds. These models also yield substantial gains in data efficiency and distillation, outperforming color-supervised teachers in student performance even when matched on classification accuracy.

Together, our findings suggest that a minimal, structure-first learning curriculum is a powerful strategy for fostering robust, efficient, and transferable vision systems—grounded in the computational principles that underlie human perception.

## 2 RELATED WORK

**Line Drawing Synthesis and Structural Representation**  Recent advances in line drawing generation have focused on balancing geometric fidelity and semantic preservation. The work by Chan et al. (2022) pioneered an unsupervised approach using geometry and semantic losses to translate photographs into sparse yet informative sketches, circumventing the need for paired datasets. Earlier methods relied on supervised training with stroke annotations or 3D geometry, limiting their applicability to arbitrary photographs (Das et al., 2020; Ganin et al., 2018; Ha & Eck, 2018; Smirnov et al., 2020). Concurrent work in unpaired translation, such as CycleGAN (Zhu et al., 2017) and UPDG (Yi et al., 2019), achieved domain adaptation but struggled with structural coherence in sparse outputs. Building on the foundations of modern line drawing synthesis from authors like Chan et al. (2022), our work leverages these semantically coherent yet sparse sketches for a novel purpose. We use these minimal representations as the basis for a learning curriculum that demonstrates robust visual understanding can be achieved by prioritizing structural information over dense, high-fidelity detail.

**Visual Pre-training**  Self-supervised pre-training has emerged as a dominant paradigm for learning robust visual representations. Methods like Masked Autoencoders (MAE) (He et al., 2021) learn by reconstructing masked patches, while contrastive and distillation-based approaches like DINO (Caron et al., 2021b) and SwAV (Caron et al., 2020) learn invariant representations by matching different views. A common thread in these methods is their reliance on dense, texture-rich color photographs as the sole training input. This paper demonstrates that a 'structure-first' approach is not mutually exclusive with these pre-training methods, but is a powerful, complementary strategy. We show that introducing line drawings into their training processes—for example, by using them as an input modality in the initial phase of frameworks like DINO and SwAV—can massively increase performance and computational efficiency, instilling a robust structural bias that enhances existing pre-training frameworks.

**Methods to Improve Shape Bias**  The texture bias of convolutional neural networks has motivated efforts to instill shape-centric reasoning. Work by Geirhos et al. (2019) showed that stylized training data improves shape sensitivity, but this did not scale well. Other approaches from Hermann et al. (2020) and Lee et al. (2022) respectively used human-like and shape-guided data augmentation techniques to induce more shape bias. Work by Dapello et al. (2023) focused on architectural improvements and induced shape bias through sparsity constraints in transformer architectures. Our work bridges these insights by using line drawings—which are naturally sparse representations—to explicitly steer models toward geometric regularization. Unlike augmentation strategies, our method leverages the inherent spatial regularity of line art to improve cross-domain generalization.

**Sparsity in Neural Representations**  Sparse coding has long been theorized as a mechanism for efficient visual processing. Recent work (Dapello et al., 2023) demonstrates that Top-K activation sparsity enhances shape bias by emphasizing part-based representations, while masked autoencoders (He et al., 2021) use spatial sparsity to improve feature learning. Line drawings naturally instantiate this principle, distilling scenes to their essential contours. Our structure-first curriculum leverages this inherent input sparsity; by beginning with an initial phase of learning on line drawings, our method encourages models to develop sparse, structure-focused representations, compelling them to discard textural noise in favor of essential invariants, a process analogous to an artist's mental decomposition of complex forms (Perdreau & Cavanagh, 2014).

## 3 METHOD

We employ a two-stage supervised learning curriculum using the STL-10 dataset with stratified sampling across data subsets ranging from 5% to 100% of the full training corpus. Our methodology compares this structure first learning approach against a baseline control where models train directly on natural color images, establishing the performance ceiling for direct supervised learning.

We convert the original STL-10 (Coates et al., 2011) training images to line drawings using the method from Chan et al. (2022), which extracts the most crucial features in line form. For the artistic style variants, we apply style transfer techniques from Huang & Belongie (2017) to generate stylized versions of the original images, creating distinct visual abstractions that test semantic transfer across diverse visual domains. We also use Canny edge detection to generate images as a naive control.

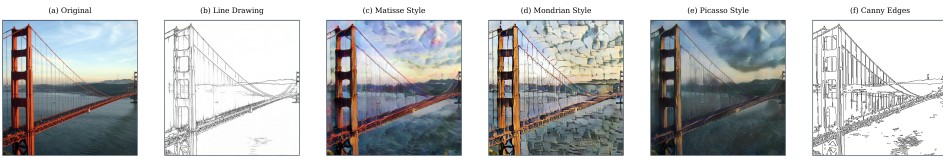

Figure 2: Different augmentations of original image

In our two stage curriculum, models first undergo a foundational training phase on source domains including line drawings and various artistic styles, all using identical class labels as the target domain. Line drawings serve as our primary focus, providing structural information without color, while the artistic styles function as arbitrary augmentations to compare against. We then train the models on natural color images, allowing them to leverage learned semantic representations while adapting to target visual characteristics.

The training configuration employs adaptive learning rates (0.001-0.01) and batch sizes (4-32) that scale with dataset size, along with early stopping (patience=10). Both training stages utilize identical data augmentation strategies including random resized cropping, horizontal flipping, and rotation. Models are evaluated on the STL10 test sets using center-cropped images, with comprehensive testing across all data percentage levels under fixed random seeds for reproducibility.

To demonstrate the broad applicability of our structure-first curriculum, we conducted extensive validation beyond the initial STL-10 experiments. We evaluated our approach across diverse architectures and datasets, training both a YOLOv8 backbone on ImageNet-1K and a Vision Transformer (ViT) on STL-10 (detailed setup provided in the appendix). Additionally, we assessed the quality and transferability of learned representations through evaluation on multiple downstream tasks, including segmentation, detection, and knowledge distillation. These comprehensive experiments

confirm that our method produces robust, generalizable features that perform well across varied computer vision applications.

# 4 RESULTS

## 4.1 BENEFITS OF A STRUCTURE-FIRST CURRICULUM

The results presented in Table 1 reveal important insights into the effects of different training curricula on model performance. Training with the two-stage curriculum of line drawings (STL10-Line) followed by color images (STL10) leads to an improvement in generalization to line drawings, with accuracy rising from 16.67% to 29.53%. At the same time, color image classification accuracy also increases from 68.79% to 73.58%, indicating that structural features learned from line drawings provide a beneficial inductive bias for learning from color representations.

| Training Strategy | Color Acc | Line Acc |
|---|---|---|
| Color Only | 68.79 | 16.67 |
| Line Only | 10.67 | 62.62 |
| Line→Color | 73.58 | 29.53 |
| Color→Line | 13.43 | 66.92 |

Table 1: Classification accuracy (%) for different training strategies.

The experiments also demonstrate a pronounced asymmetry in performance depending on the order of training modalities. When models are trained first on color images and then on line drawings, color image classification accuracy drops sharply to 13.43%, despite strong performance on line drawings (66.92%). Conversely, training first on line drawings and then on color images results in high color image accuracy (73.58%) and moderate line drawing accuracy (29.53%). This suggests that features derived from line drawings are more generalizable and facilitate adaptation to color images, whereas color features are more specific and less transferable to line drawings.

These findings have practical implications for the design of image learning systems. A structure-first curriculum offers a robust initialization that is particularly effective when computational resources are limited. The results also highlight the risk of catastrophic forgetting when adapting models trained solely on color images to line drawings. Overall, the results support the notion that structural features learned from line drawings are more transferable and beneficial for generalization across different visual modalities.

| Training Strategy | STL10 Acc. |
|---|---|
| Color Only | 55.04 |
| Line → Color | 60.97 |

| Training Strategy | ImageNet-1k Acc. |
|---|---|
| Color Only | 67.87 |
| Line → Color | 68.82 |

Table 2: STL10 classification accuracy (%) for ViT-Tiny (5.7M parameters) architecture under different training strategies.

Table 3: ImageNet-1k classification accuracy (%) for YOLO-v8 backbone (11M parameters) under different training strategies.

To validate that our findings are not limited to a single architectural class, we conducted additional experiments with both transformer-based and detection-oriented models. As shown in Table 2, applying our structure-first curriculum to a ViT-Tiny architecture on the STL10 dataset yields a substantial performance increase, boosting accuracy from 55.04% to 60.97%. This demonstrates that the benefits of structural learning are highly effective for modern transformer models.

Furthermore, to test our approach on different model types and at a larger scale, we used a YOLO-v8 backbone on the ImageNet-1k dataset. The results in Table 3 show a consistent, albeit more modest, improvement from 67.87% to 68.82%. This confirms that the positive effects of our curriculum extend to modern detection-based backbones and scale to more complex, large-scale datasets.

Notably, these results corroborate our central hypothesis that an initial learning phase on line drawings induces beneficial inductive biases that transfer across visual domains. The convergent evidence from convolutional (ResNet-18, YOLO-v8) and transformer-based (ViT-Tiny) architectures

establishes this structure-first approach as a broadly applicable technique for improving visual representation learning.

## 4.2 DATA EFFICIENCY

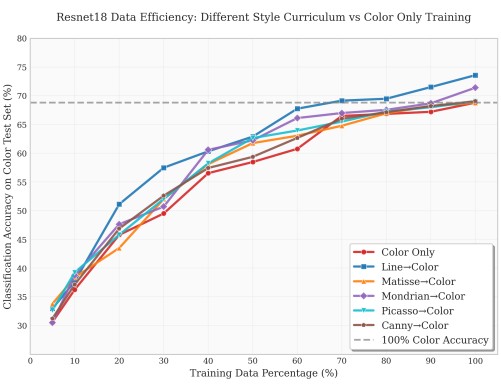 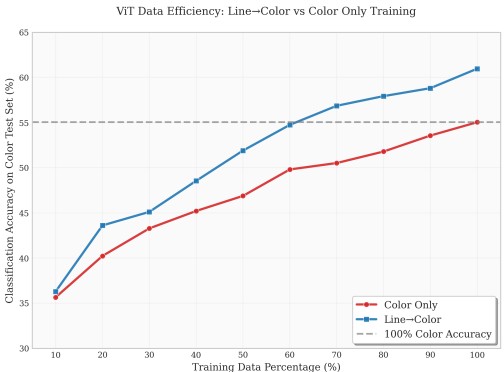

Figure 3: ResNet-18 data efficiency results of training with various augmentations of STL10 and finetuning on the base STL10 data.

Figure 4: ViT-Tiny data efficiency results of training on STL10 with our Line→Color curriculum vs Color only

Our experimental results provide compelling evidence for the data efficiency advantages of line drawing curriculum learning over conventional color-only training approaches. As illustrated in Figure 3, the Line-Color method demonstrates superior learning efficiency across all training data regimes, achieving the baseline color accuracy (69%) with only 70% of the training data—representing a substantial 30% reduction in labeled data requirements. This efficiency gain manifests early in the training process, with the Line-Color approach consistently outperforming the color-only baseline by 3-5 percentage points at equivalent data volumes. Notably, the learning trajectory exhibits a steeper ascent during the initial training phases, suggesting that the line drawing curriculum provides a more structured optimization landscape that facilitates faster convergence to target performance levels.

The comparative analysis with style controls (Matisse-Color, Mondrian-Color, and Picasso-Color, Canny-Color) further validates the specificity of line drawing advantages. We specifically include Canny edge images as a naive control, as they represent a classic, low-level approach to structure extraction that lacks the semantic coherence of learned line drawings. While some of these alternative visual abstractions yield improvements over the color-only baseline, they are significantly less effective than line drawings in achieving data efficiency. The other methods require substantially more training data to reach baseline performance compared to the line drawing approach. This differential performance strongly supports our theoretical framework that line drawings tap into fundamental shape and topology processing mechanisms inherent to visual recognition systems. The reduced effectiveness of alternative abstractions demonstrates that the efficiency gains are not merely artifacts of stylistic variation, but rather reflect the unique capacity of line drawings to provide structured, shape-preserving visual representations that align with the computational principles underlying human visual processing.

Furthermore, these data efficiency benefits extend beyond convolutional architectures. As demonstrated with a Vision Transformer (ViT-Tiny) in Figure 4, the advantages of the structure-first curriculum are even more pronounced. The ViT model trained with the Line-Color curriculum achieves the final performance of the color-only baseline while using approximately 40% less training data, a greater reduction than observed with ResNet-18. This consistent outperformance across both CNN and transformer-based models underscores the general applicability of our approach, confirming that the induced structural priors provide a robust, cross-architecture benefit for efficient visual learning.

## 4.3 COMPLEMENTING EXISTING PRETRAINING METHODS

To demonstrate that our structure-first approach is not merely an alternative to pre-training but can actively enhance it, this work tests its impact on powerful, modern pre-training paradigms. The

investigation explores whether using line drawings as an input modality during the pre-training phase can improve performance and efficiency.

As case studies, our 'structure-first' phase is integrated with DINO (Caron et al., 2021a) and SwAV (Caron et al., 2020), two state-of-the-art pre-training methods. All experiments are implemented using the `solo-learn` library (da Costa et al., 2022). A strong baseline was established for each method by training a ResNet-18 model for 400 epochs on color images from the ImageNet-100 dataset. These baselines are compared against mixed-modality curricula that use a matched (400) or reduced (250) total epoch budget. The results are presented in Table 4.

| Method | Line Epochs | Color Epochs | Total Epochs | Top-1 (%) | Top-5 (%) |
|---|---|---|---|---|---|
| DINO (Baseline) | 0 | 400 | 400 | 74.92 | 92.78 |
| DINO (Ours) | 150 | 250 | 400 | 77.22 (+2.30) | 94.18 (+1.40) |
| DINO (Ours) | 150 | 100 | 250 | 75.54 (+0.62) | 93.18 (+0.40) |
| SwAV (Baseline) | 0 | 400 | 400 | 74.28 | 92.84 |
| SwAV (Ours) | 150 | 250 | 400 | 77.62 (+3.34) | 94.34 (+1.50) |
| SwAV (Ours) | 150 | 100 | 250 | 74.40 (+0.12) | 92.96 (+0.12) |

Table 4: Performance of DINO and SwAV baselines versus our mixed-modality curriculum on ImageNet-100 (ResNet-18). Our structure-first phase significantly boosts performance with matched compute and achieves superior or comparable results with a 37.5% compute reduction.

The results from these experiments yield several critical insights. Our method provides significant gains across different pre-training frameworks. With an identical 400-epoch compute budget, our curriculum boosts DINO's Top-1 accuracy by +2.30% and SwAV's by an even larger +3.34%. This demonstrates that the performance gain is not an artifact of a larger compute budget; rather, introducing structural data as an initial input modality enhances the final representation.

The benefits are also clear in the reduced-compute scenarios. With a drastically reduced 250-epoch budget (37.5% less compute), our curriculum still outperforms both 400-epoch baselines. The DINO (150+100) model achieves 75.54% Top-1, surpassing its baseline (74.92%), and the SwAV (150+100) model achieves 74.40%, slightly edging out its own 400-epoch baseline (74.28%). These case studies confirm that our structure-first approach and pre-training are not mutually exclusive. Using line drawings as an input modality is a complementary strategy that can effectively supplement existing pre-training methods, improving both their final performance and their computational efficiency.

### 4.4 MORE FOCUSED ATTENTION

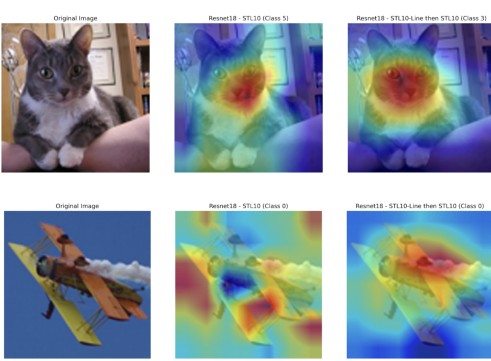 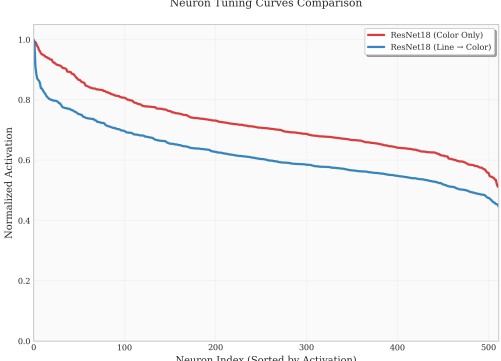

Figure 5: GradCAM visualizations comparing attention heat maps for models trained on color images vs trained with our curriculum.

Figure 6: Neuron tuning curves of base ResNet-18 vs ResNet-18 with our curriculum. (Normalized activations over 500 test images)

To understand the mechanisms driving these performance improvements, we analyzed the attention patterns of models trained under different learning curricula. Figure 5 presents GradCAM (Gildenblat & contributors, 2021) visualizations comparing models trained exclusively on color images versus those trained with our structure-first curriculum (STL10-Line followed by STL10 color).

Models initialized with structure-first learning on line drawings demonstrate markedly more focused attention patterns, with heat maps concentrated on the primary subject. For the cat image, the structure-first model concentrates on facial features and body contours, while the color-only model's activation is more diffuse. Similarly, for the airplane, the structure-first model focuses on the aircraft's structural elements, unlike the scattered activation of the color-only model. A more quantitative analysis of these GradCAM activations is included in the supplementary material.

## 4.5 SPARSE NEURONAL TUNING CURVES

The neuron tuning curve presented in Figure 6 provides quantitative support for these qualitative observations. We evaluate the activation tuning curve for the last layer of the trained ResNet-18. Each neuron in Figure 6 represents one channel. We take the average firing activations from each channel and normalize them according to their maximum firing activation within each setting. The resulting activation tuning curve shows that ResNet-18 first trained on line drawings and then adapted to natural images (blue curve) exhibits a much sharper tuning curve in activation compared to the baseline counterpart (red curve). This pattern indicates that strong responses are concentrated among a small subset of neurons, while most units remain weakly or sparsely engaged. Such a rapid fall-off shows an increased sparsity and efficiency in neural representations.

Instead of distributing activation broadly, the network from the structure-first curriculum allocates capacity to the most informative neurons, silencing irrelevant or noisy features. The base model, by contrast, shows a more gradual decrease, activating a broader collection of neurons, including many that likely encode non-discriminative or redundant information.

This selective focus echoes principles of efficient coding: the model learns to emphasize features that matter for object recognition while suppressing spurious or noisy activations. This transition to more selective, quieter representations improves performance without any trade-off. The structure-first model achieves stronger accuracy on both minimal and natural images.

These results encapsulate the representational transformation induced by an initial phase of minimal, structural learning, which encourages the model to specialize its detectors for geometric cues.

## 4.6 INCREASED SHAPE BIAS

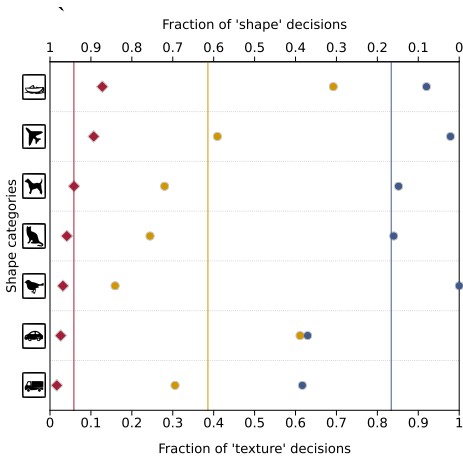

Figure 7: Shape versus texture decision analysis across object categories. Red diamonds indicating human comparison data from lab experiments, yellow circles indicate ResNet-18 Line→Color, and blue circles represent base ResNet-18 results.

To further assess the inductive biases, we evaluated shape versus texture preferences using the model-vs-human benchmark (Geirhos et al., 2021). This benchmark presents cue-conflict images in which shape and texture cues are intentionally mismatched, allowing for a direct comparison of classification strategies between models and human observers.

Figure 7 displays the distribution of shape and texture decisions across object categories (these are the 7 categories in the cue-conflict dataset that coincide with STL10 classes). Models first trained on line drawings and subsequently adapted to color images demonstrate a pronounced shift toward shape-based decisions, as indicated by the clustering of yellow circles toward higher fractions of shape choices. In contrast, models trained only on natural images show a greater reliance on texture cues, with more blue circles distributed toward the texture-dominant end of the spectrum. This pattern suggests that an initial phase of learning on line drawings encourages the model to prioritize global structural features over local textural information when resolving ambiguous visual input.

These results indicate that our structure-first approach not only enhances classification performance and attention focus, but also aligns model decision-making more closely with human perceptual strategies. This increased shape bias likely contributes to the improved robustness and generalization observed in downstream tasks, as models become less reliant on superficial texture cues and more attuned to the underlying structure of visual objects.

### 4.7 BENEFITS IN SEMANTIC SEGMENTATION

We further show that an initial phase of learning on line drawings enhances generalization to dense prediction tasks such as semantic segmentation. To evaluate this, we adapt ResNet-18 models—initialized either from color-only training or from our structure-first curriculum—on the ADE20K benchmark.

As shown in Figure 8, the model initialized with line drawings consistently outperforms the color-only baseline across multiple semantic categories. The improvement is especially pronounced for object-centric classes (e.g., "chair", "car"), where shape plays a dominant role in visual discrimination. This demonstrates that structural priors learned from minimal representations translate effectively to tasks requiring precise spatial reasoning.

These results extend our findings beyond classification and detection, highlighting that our structure-first curriculum induces representations that generalize across both sparse and dense vision tasks.

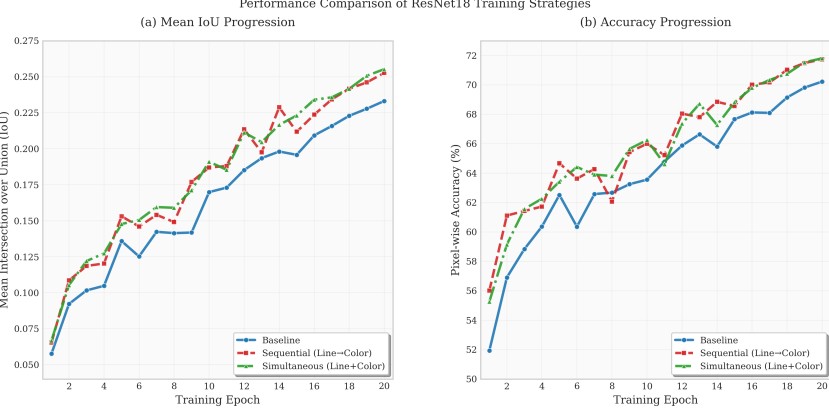

Figure 8: Different learning curricula on ResNet-18 generalizing to segmentation on ADE20K.

### 4.8 BENEFITS IN OBJECT DETECTION

We also run experiments similar to those shown in Figure 3, but on the object detection task using the PASCAL VOC dataset. Figure 9 illustrates that models trained with our structure-first curriculum consistently outperform color-only models across all training data regimes. Notably, our Linedrawing→Photo (Ours) model trained with just 90% of the data achieves an mAP of 56.74, surpassing the Photo Only model trained with the full 100% dataset, which reaches 56.14. This highlights a key advantage of our approach: better performance with less data. The performance gap is particularly

prominent in the low-data regime—for example, at 10% training data, our model achieves 22.16 mAP compared to 18.17 for the color-only baseline. These findings suggest that an initial learning phase on structure-focused line drawings enables models to extract more efficient and transferable features for object detection, leading to significant gains in data efficiency and generalization.

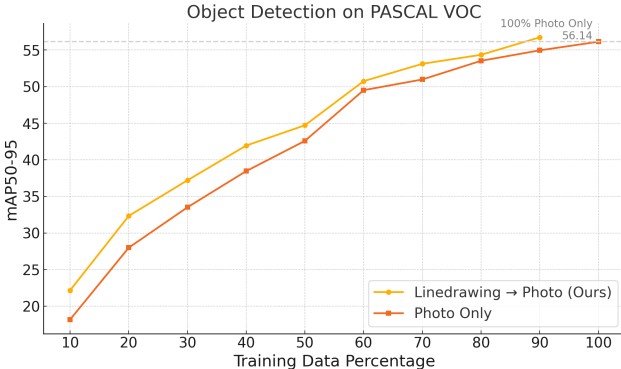

Figure 9: Improved Data Efficiency with Linedrawing→Color Training on PASCAL VOC for Object Detection.

## 4.9 COMPACT AND TRANSFERABLE REPRESENTATIONS

One of the core advantages of a structure-first curriculum is the emergence of compact and structurally focused internal representations. To investigate this, we first analyze the intrinsic dimensionality of neural activations in a ResNet-18 model trained under two regimes: color-only vs. linedrawing → color. We extract penultimate layer activations across STL10 test images and compute principal components (PCs) via PCA.

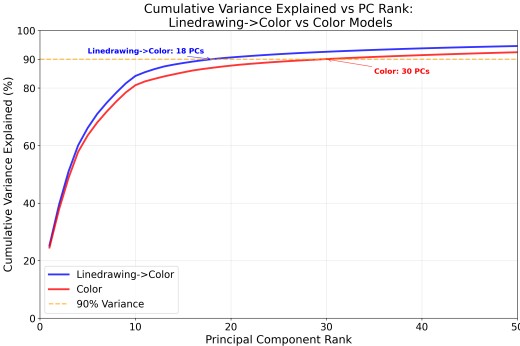

Figure 10: The structure-first Linedrawing→Color model learns a lower-dimensional representational manifold. This is demonstrated by performing Principal Component Analysis (PCA) on the model's $\mathbb{R}^{512}$ latent space activations for the STL10 test set. To capture 90% of the variance, the structure-first model requires only 18 PCs, whereas a standard color-only model needs 30 PCs, indicating a more compact representation.

As shown in Figure 10, the model from the structure-first curriculum requires only 18 PCs to explain 90% of the variance, compared to 30 for the color-only model—suggesting that the model organizes information more economically and filters out redundant features. This compressed manifold structure aligns with principles of efficient coding and complements earlier findings on activation sparsity (Figure 6).

We further examine whether this structural compactness benefits model distillation. Using a standard teacher-student framework, we compare the performance of student models (MobileNetV1, ResNet-8, VGG-8) distilled from ResNet-18 teachers trained with either color-only or line-drawing → color supervision. As shown in Table 5, students distilled from teachers developed with the structure-first curriculum consistently outperform those from color-only teachers. For example, a

| Teacher | Teacher Acc | Student | Distillation Acc |
|---|---|---|---|
| Linedrawing→Color (1) | 73.58 | MobileNetV1 | **77.06** |
| Linedrawing→Color (2) | 69.01 | MobileNetV1 | 75.48 |
| Color (3) | 68.89 | MobileNetV1 | 73.83 |
| Linedrawing→Color (1) | 73.58 | ResNet-8 | **72.75** |
| Linedrawing→Color (2) | 69.01 | ResNet-8 | 71.08 |
| Color (3) | 68.89 | ResNet-8 | 68.81 |
| Linedrawing→Color (1) | 73.58 | VGG-8 | **77.25** |
| Linedrawing→Color (2) | 69.01 | VGG-8 | 75.41 |
| Color (3) | 68.89 | VGG-8 | 73.15 |

Table 5: This table compares knowledge distillation from a line-drawing → color teacher (2) and a standard color teacher (3). For a fair comparison, their accuracies are matched by training (2) on 70% of the data while (3) uses 100%. The line-drawing teacher (2) yields superior student models. For reference, we also show the performance of an unmatched, higher-performing line-drawing teacher (1), which was trained on 100% of the data.

MobileNetV1 distilled from a teacher from the structure-first curriculum (trained with 70%-data to reduce the confounding factor of a stronger teacher) achieves 75.48% accuracy—exceeding the 73.83% achieved by the color-only teacher while the two teachers share almost the same object recognition performance.

These results suggest that the structure-first curriculum not only reduces the complexity of internal representations, but also produces knowledge that is easier to transfer, yielding more performant student models.

## 5 CONCLUSION

Our findings show that a structure-first curriculum using line drawings induces compact, structure-aware representations that generalize across tasks and enable more effective distillation into lightweight models. This structure-first learning approach improves performance, enhances data efficiency, and yields representations that are easier to compress and transfer. More broadly, our results offer a computational lens on the longstanding observation that humans can easily understand line drawings—suggesting that prioritizing geometry over appearance may support more general and efficient visual reasoning, both in artificial and biological systems.

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

# A APPENDIX

## A.1 SUPERVISED LEARNING STATISTICAL ANALYSIS

| Data (%) | Color Only | Line→Color | Matisse→Color | Mondrian→Color | Picasso→Color |
|---|---|---|---|---|---|
| 5 | $30.62 \pm 0.77\%$ | $32.95 \pm 0.59\%$ | $33.79 \pm 0.92\%$ | $30.49 \pm 1.87\%$ | $32.64 \pm 0.68\%$ |
| 10 | $36.25 \pm 1.66\%$ | $37.68 \pm 0.30\%$ | $38.40 \pm 2.34\%$ | $38.76 \pm 0.94\%$ | $39.25 \pm 1.56\%$ |
| 20 | $45.78 \pm 3.22\%$ | $51.12 \pm 2.96\%$ | $43.50 \pm 1.73\%$ | $47.64 \pm 2.48\%$ | $45.80 \pm 3.15\%$ |
| 30 | $49.51 \pm 4.10\%$ | $57.48 \pm 1.08\%$ | $52.01 \pm 3.67\%$ | $50.70 \pm 1.42\%$ | $52.04 \pm 2.19\%$ |
| 40 | $56.53 \pm 0.75\%$ | $60.33 \pm 1.58\%$ | $58.10 \pm 0.84\%$ | $60.59 \pm 2.73\%$ | $58.24 \pm 1.29\%$ |
| 50 | $58.47 \pm 1.27\%$ | $62.89 \pm 1.60\%$ | $61.76 \pm 2.05\%$ | $62.24 \pm 0.76\%$ | $62.64 \pm 2.81\%$ |
| 60 | $60.76 \pm 1.87\%$ | $67.75 \pm 1.15\%$ | $63.04 \pm 1.38\%$ | $66.12 \pm 2.94\%$ | $63.95 \pm 0.89\%$ |
| 70 | $66.49 \pm 0.71\%$ | $69.15 \pm 4.03\%$ | $64.78 \pm 2.67\%$ | $66.99 \pm 1.03\%$ | $65.50 \pm 1.85\%$ |
| 80 | $66.84 \pm 2.09\%$ | $69.48 \pm 2.13\%$ | $66.94 \pm 0.95\%$ | $67.55 \pm 1.74\%$ | $67.24 \pm 2.36\%$ |
| 90 | $67.21 \pm 2.34\%$ | $71.51 \pm 0.46\%$ | $68.19 \pm 3.08\%$ | $68.68 \pm 0.62\%$ | $68.00 \pm 1.47\%$ |
| 100 | $68.79 \pm 0.91\%$ | $73.58 \pm 0.42\%$ | $68.77 \pm 1.53\%$ | $71.40 \pm 0.56\%$ | $69.03 \pm 0.77\%$ |

Table 6: Aggregated results with std dev across all seeds showing performance comparison between Color Only training and different initial learning modalities.

To rigorously evaluate the statistical significance of our findings, we conducted Wilcoxon signed-rank tests comparing each initial learning modality against the Color Only baseline. The Wilcoxon signed-rank test is a non-parametric statistical test that assesses whether there is a significant difference between paired observations without assuming a normal distribution, making it particularly appropriate for our experimental setup where we have matched pairs of performance measurements across different data percentages.

For each comparison, we tested the null hypothesis $H_0$: median difference = 0 against the alternative hypothesis $H_1$: method performance > Color Only baseline (one-tailed test). The test was applied to 11 paired observations corresponding to the performance differences at each data percentage level (5%, 10%, 20%, ..., 100%).

Statistical Results: Line→Color vs Color Only shows a median improvement of +4.33 percentage points (W-statistic: 66, p-value: 0.002, highly significant), with Line→Color outperforming Color Only in all 11 data conditions (11/11 wins). Other stylistic augmentations showed: Mondrian→Color +1.47 percentage points (p = 0.037, significant), Picasso→Color +0.24 percentage points (p = 0.195, not significant), and Matisse→Color -0.02 percentage points (p = 0.500, not significant).

Implications: The statistical analysis provides strong evidence for several key conclusions. The Line→Color approach demonstrates highly significant improvement over the Color Only baseline (p = 0.002), with consistent gains across all tested data percentages. This statistical significance, combined with the perfect win rate (11/11), indicates that the performance benefit is both reliable and substantial. Critically, our results show that Line→Color's benefits cannot be attributed merely to stylistic augmentation effects. While Mondrian→Color shows modest improvement (+1.47%, p = 0.037), both Matisse→Color and Picasso→Color fail to achieve statistical significance. The substantial difference between Line→Color's improvement (+4.33%) and that of other stylistic methods suggests that line drawings possess unique structural properties that facilitate more effective learning. The consistent performance gains across all data percentages (5% to 100%) demonstrate that an initial phase of learning on line drawings provides benefits in both low-data and high-data scenarios. The median improvement of 4.33 percentage points represents a practically meaningful enhancement in classification performance that could translate to significant real-world benefits. These statistical findings support our central hypothesis that line drawings provide structurally advanta-

geous representations for subsequent learning phases, offering benefits that extend beyond those achievable through general stylistic augmentation approaches.

## A.2 COMPUTING INFRASTRUCTURE

Experiments are run on a computing cluster equipped with dual NVIDIA L40 GPUs and AMD EPYC 7443 24-Core Processors running at 2.84GHz. Each experimental job utilized 8 CPU cores and 32GB of system RAM. The computing environment uses Springdale Open Enterprise Linux 8.6 (Modena) operating system on x86-64 architecture.

## A.3 DATASETS PREPARATION

To generate the line drawings for our initial structural learning phase, the pre-existing "Anime style" model from `https://github.com/carolineec/informative-drawings` (Chan et al., 2022) was used to create line drawing versions of STL10 (Coates et al., 2011) and ImageNet-1K (Russakovsky et al., 2015).

For the other augmentations, code from `https://github.com/xunhuang1995/AdaIN-style` (Huang & Belongie, 2017) was used to generate Matisse, Mondrian, and Picasso style STL10 images with degree of stylization $\alpha = 0.6$.

The base ADE20K (Zhou et al., 2017) and Pascal VOC (Everingham et al., 2010) are also used downloaded for evaluation on segmentation and detection.

## A.4 SUPERVISED LEARNING EXPERIMENTS

We conduct experiments comparing two training approaches: a baseline Color-Only Training (direct training on STL10 color images) and our Line→Color curriculum (an initial phase of learning on line drawings followed by a second phase on color images). As a control, we replicated this same curriculum with 3 different arbitrary augmentations (Matisse, Mondrian, Picasso) to show that performance benefits came from the intrinsic properties of line drawings and not just the effects of using an alternative visual modality for an initial learning stage. The dataset is STL10 with 10-class image classification, using 80% train / 20% validation split with stratified sampling. We test data percentages from 5% to 100% in increments of 5% or 10%.

The model architecture uses ResNet-18 trained from scratch, modified with a final linear layer (512 $\rightarrow$ 10) for 10-class classification, totaling approximately 11.2M parameters.

During hyperparameter development, we explored learning rates from 0.0001 to 0.01, batch sizes from 4 to 64, early stopping patience from 5 to 15 epochs, and weight decay from 1e-5 to 1e-3. Final hyperparameters use adaptive learning rates and batch sizes based on dataset size: 0.001/4 for small datasets ($\leq$300 samples), 0.003/8 for medium datasets ($\leq$1000 samples), 0.005/16 for larger datasets ($\leq$2000 samples), and 0.01/32 for full datasets ($>$2000 samples).

Training configuration uses SGD optimizer with momentum 0.9, ReduceLROnPlateau scheduler with patience 3, early stopping with patience 10 epochs, and maximum 100 epochs. The primary evaluation metric is test accuracy on held-out test sets.

All experiment runs use system-generated randomness with no fixed seeds, reflecting natural variation and real-world training variability. Experimental runs use 5 runs per configuration with different random seeds. The training code is available in the code appendix in `Training/train_resnet18.py`.

We also conducted experiments using Vision Transformer Tiny (ViT-Tiny) architecture for comparison with the ResNet-18 baseline. The model uses the vit_tiny_patch16_224 configuration from the timm library, trained from scratch on STL10 with 10-class classification, containing approximately 5.7M parameters with patch size 16x16 and input resolution 224x224. The architecture includes 12 transformer layers, 3 attention heads, hidden dimension 192, learnable position embeddings, and a classification head (192 $\rightarrow$ 10) for 10-class classification.

The training configuration was optimized specifically for small datasets based on recommendations from (Steiner et al., 2022). We used AdamW optimizer with learning rate 0.0005, weight decay 0.01,

and betas (0.9, 0.999), combined with cosine annealing learning rate schedule and 10% warmup epochs. The batch size was set to 32 (reduced from standard 64 for better gradient estimates on small datasets) with light augmentation strategy including RandomResizedCrop (scale 0.85-1.0), RandomHorizontalFlip (p=0.5), and mild ColorJitter. Regularization included conservative mixup ($\alpha = 0.1$, applied 30% of the time), label smoothing (0.05), and gradient clipping (max_norm=0.5). Training used maximum 150 epochs with early stopping patience of 20 epochs.

The configuration specifically addresses challenges of training ViTs on small datasets through reduced regularization, conservative augmentation, smaller batch sizes, and extended patience periods. The curriculum learning approach follows the same Line→Color transfer strategy as the ResNet-18 experiments, with Xavier uniform initialization for linear layers and standard initialization for Layer-Norm components following best practices for training ViTs from scratch. Training code is available in the code appendix in `Training/train_tiny_vit.py`

### A.5 GradCAM Distribution Analysis

We analyze the spatial distribution of model attention by counting distinct high-activation regions and average size of said regions in GradCAM heatmaps. This quantifies whether models focus on concentrated areas or distribute attention across multiple regions.

For each GradCAM heatmap, we compute a dynamic threshold as the 85th percentile of heatmap values. We then create a binary mask where pixels above the threshold are marked as attention regions. To remove noise, we filter out regions smaller than 10 pixels and count distinct connected components using 8-connectivity labeling.

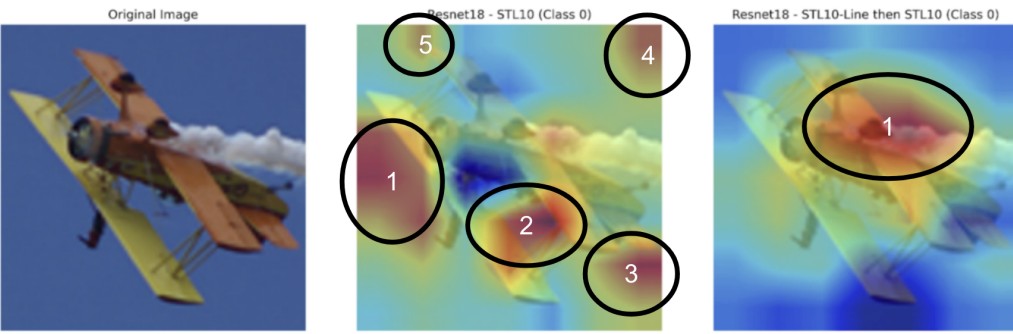

Figure 11: Example of counting number of high activation regions from GradCam. The GradCAM heatmap for base ResNet-18 has 5 distinct activation regions on this image while the ResNet-18 from the structure-first curriculum has one larger focused activation region.

The attention scatter metric is defined as the number of distinct attention regions per image. We visualize the distribution of region counts as a histogram to show the frequency of different attention scatter patterns.

For mean region size, we calculate the area of each connected component in pixels and compute the average region size per image. This metric indicates whether attention regions tend to be large and concentrated or small and scattered. Quantitative analysis confirms that the structure-first curriculum leads to a larger number of images with a single, compact attention region, whereas color-only training results in more fragmented attention (Figure 12). The GradCam analysis is available in the code appendix in `Evaluation/gradcam_activations.py`.

### A.6 Neuron Tuning Curve Analysis

The neuron tuning curve analysis examines how different initial learning modalities affect the activation patterns of individual channels in the final convolutional layer of ResNet-18 models. In this analysis, each "neuron" corresponds to one of the 512 channels in the final convolutional layer (layer4) of ResNet-18. The activation of each neuron is computed by averaging the spatial dimensions (height and width) of the corresponding feature map.

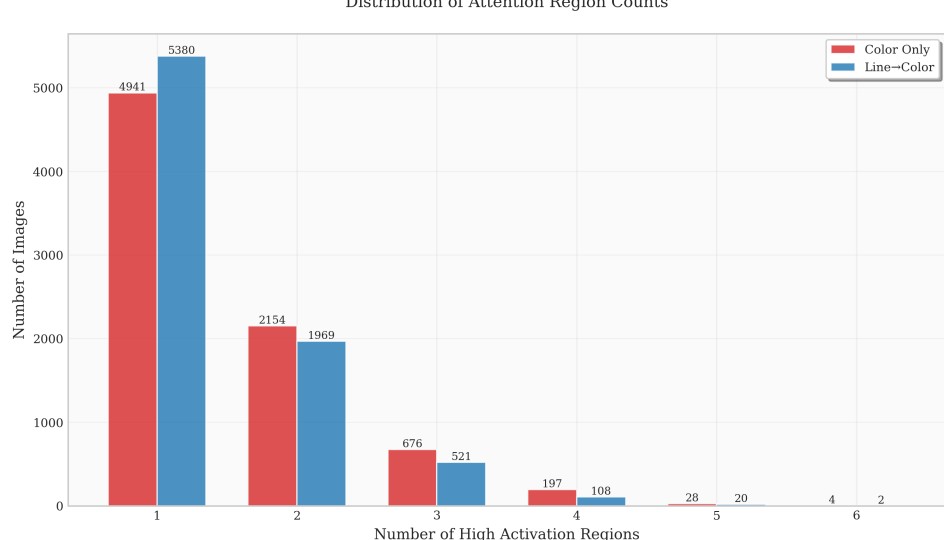

Figure 12: Number of distinct high activation regions across all STL10 Test images. High activation regions are marked as the number of regions above the 85th percentile in GradCam activation.

The analysis processes a dataset of test images (up to 500 images) through both models: ResNet-18 trained on color images only (Color Only), and ResNet-18 first trained on line drawings followed by a second phase of training on color images (Line→Color). For each image, we extract the 512-dimensional neuron activation vector using forward hooks registered on the layer4 module.

For each model, we compute the mean activation across all processed images for each of the 512 neurons. The neurons are then sorted by their mean activation values in descending order, creating a tuning curve that shows the distribution of neuron response strengths. The curves are normalized by dividing by the maximum activation value to facilitate comparison between models.

The resulting tuning curves reveal the degree of specialization in the learned representations. A steeper, more concentrated curve indicates that fewer neurons are highly active, suggesting more specialized, sparse representations. By comparing the curves between Color Only and Line→Color models, we can assess whether an initial phase of learning on line drawings leads to more efficient, specialized neuronal representations. The code for this is available in the code appendix in `Evaluation/neuron_tuning_curves.py`.

## A.7 GENERALIZING TO SEGMENTATION

To evaluate how well ResNet-18 models initialized with different visual modalities generalize to semantic segmentation, we conducted experiments on the ADE20K dataset using a standard encoder-decoder architecture. We compared two ResNet-18 encoders developed through different curricula: one trained on color images and another trained sequentially on line drawings followed by color images (Line→Color). We also include another ResNet-18 that is simultaneously trained on line and color images to investigate whether training order affects generalization ability.

We modified the semantic segmentation framework from https://github.com/CSAILVision/semantic-segmentation-pytorch (Zhou et al., 2018) to support our custom encoders. The training configurations closely match the original framework's settings to ensure a fair comparison. We employed a Pyramid Pooling Module (PPM) decoder architecture with ResNet-18 as the encoder backbone.

The training setup was identical for all models to ensure fair comparison. We trained for 20 epochs with a batch size of 2 per GPU using SGD optimization. The learning rate was set to 0.02 for both encoder and decoder, with a polynomial decay schedule ($lr_{current} = lr_{initial} \times (1 - \frac{iter}{max\_iter})^{0.9}$).

Both the encoder and decoder were trained end-to-end, allowing the learned representations to be adapted for the segmentation task.

The models were trained on ADE20K with 150 semantic classes, using images resized to multiple scales $(300, 375, 450, 525, 600)$ with a maximum size of 1000 pixels. We employed standard data augmentation including random horizontal flipping and used the negative log-likelihood loss with ignored index -1 for handling invalid labels.

Both models were evaluated using the same checkpoint at epoch 20. The evaluation metrics include pixel accuracy and mean intersection-over-union (mIoU) on the ADE20K validation set. This experimental setup allows us to directly compare the representational quality of the two learning curricula by measuring their performance on a downstream semantic segmentation task. The code for this is available in the code appendix in `Training/Segmentation`.

## A.8  OBJECT DETECTION IN PASCAL VOC

We utilize the PASCAL VOC dataset to evaluate data efficiency on object detection. For our curriculum, we first train the model on line drawings for 200 epochs and then train on color images for another 200 epochs. The base model is trained on photographic images alone for 200 epochs. We follow the training setup from ultralytics Reis et al. (2023) and use the YOLO-v8-n architecture. The code to run experiments in this section is located under `Training/ObjectDetection`.

## A.9  KNOWLEDGE DISTILLATION EXPERIMENTS

We utilize the STL10 classification task for knowledge distillation. The codebase is included in the supplementary folder `Training/KnowledgeDistillation` as well as the experiment setup. The experiments used three types of teachers: (1) The best performing model developed through our Line→Color curriculum; (2) The model developed with the Line→Color curriculum, where both learning stages are conducted using only 70% of the data. With less data, the model performance drops but still matches the baseline training with raw color images on 100% data. Using such a teacher ensures a fair comparison between the model from our structure-first curriculum and the baseline model. The experimental results in Table 5 in the main manuscript suggest that the line drawings produce a better representation that is superior when used for knowledge distillation. (3) The baseline model trained on 100% of the training data.

## A.10  CODE

Source code for this project can be found at the following anonymized repository: `https://anonymous.4open.science/r/ICLR2026-Line-Drawing-FA81`.

