# OpenReview forum: "Learning More by Seeing Less: Structure-First Learning for Efficient, Transferable, and Human-Aligned Vision"
_ICLR.cc/2026/Conference — Submitted to ICLR 2026_

### Official Review · Reviewer_4TRq · 2025-10-23

**Soundness:** 1
**Presentation:** 2
**Contribution:** 2
**Rating:** 2
**Confidence:** 4

**Summary:**

This work presents a structure-first learning paradigm that uses line drawing as the initial training modality to induce more compact and generalizable visual representations. The approach is motivated from the perspective of how humans learn, and the paper argues that structure is a key element that should be focused on instead of relying on unstable color or texture. The paper investigates several task where a model is first trained on line drawing and later trained on the color images. Experiments investigate the attention of the model, data efficiency, and the intrinsic dimensionality of the representations.

**Strengths:**

1. A nicely structured and cleanly written manuscript.
2. Nicely presented figures and tables.
3. An interesting topic with high relevance to the computer vision domain.

**Weaknesses:**

1. The experimental evaluation is limited.

(a) The proposed approach of training on line drawings before training the color images is compared with training solely on the color images. But it is widely established that pretraining is highly beneficial in computer vision [1, 2, 6], and a range of method exists to do this kind of pretraining. Most notably, self-supervised learning methods have shown great performance and is almost the standard choice to perform pretraining in computer vision [1, 2, 3]. And if the focus here is more on curriculum learning, that is also a field with a wide range of established methods [5, 6]. The core issue here is that without any comparison to relevant baselines, it is unclear if the propose structure-first learning scheme that is beneficial or just pretraining.

(b) Furthermore, the evaluation of the attention of the model is poorly motivated and unclear. The qualitative experiments in Figure 5 is not enough to support the claims in the paper, and the quantitative results in the supplementary are also unclear. There are already established protocols for evaluating such heatmaps [8], which would give much more insight into the potential benefit on the proposed training paradigm.

(c) Then, the description of the experimental setup is not detailed and it is unclear how many of the experiments are setup. My understanding of the experiments is that a model is first trained on the line drawings and the then trained further on the color images. That is compared to a model trained from scratch on the color images. If that understanding is correct, the baselines will have trained for a shorter time than the proposed methods. Looking at Figure 8, it seems like performance is still improving, and it is unclear what will happen if all models are left to complete the training. In light of [7], this seems to resemble many of the findings on curriculum learning and its potential benefits.

(d) Lastly, performance should be evaluated across numerous training runs and statistical analysis for significance should be performed.

2. Related work does not focus on relevant literature. The 3 paragraphs in the related work described prior works related to "Line Drawing Synthesis and Structural Representation", "Methods to Improve Shape Bias", and "Sparsity in Neural Representations". But the core idea of this paper is representation learning a method for pretraining. In my opinion, the related work should be focused on all the recent advances in representation learning through e.g. self-supervised learning. That would also allow the reader to understand the position of the paper and what would be natural baselines to compare it with. Also, curriculum learning should be discussed and described in the related work.

- [1] Chen et al., A Simple Framework for Contrastive Learning of Visual Representations, ICML 2020
- [2] He et al., Masked Autoencoders Are Scalable Vision Learners, CVPR, 2022
- [3] Assran et al., Self-Supervised Learning from Images with a Joint-Embedding Predictive Architecture, CVPR, 2023
- [4] Vincent et al., Extracting and composing robust features with denoising autoencoders, ICML 2008
- [5] Wang et al., EfficientTrain++: Generalized Curriculum Learning for Efficient Visual Backbone Training, TMPAMI, 2024.
- [6] Soviany et al., Curriculum Learning: A Survey, IJCV 2022
- [7] Wu et al., WHEN DO CURRICULA WORK?, ICLR 2021
- [8] Hedström et al., Quantus: An Explainable AI Toolkit for Responsible Evaluation of Neural Network Explanations and Beyond, JMLR 2023

**Questions:**

1. How does the structure-first paradigm compare to standard self-supervised pretraining methods such as contrastive learning [1], masked modeling [2], joint predictive architectures [3], or just simply autoencoder pretraining [4]?
2. How does the structure-first paradigm compare to established curriculum learning algorithms [6], for example like [5].
3. Explain how the findings in this work should be interpreted in light of [7]?
4. How does a quantitative comparison scores of the attention maps look with more established metrics like presented in [8]?

- [1] Chen et al., A Simple Framework for Contrastive Learning of Visual Representations, ICML 2020
- [2] He et al., Masked Autoencoders Are Scalable Vision Learners, CVPR, 2022
- [3] Assran et al., Self-Supervised Learning from Images with a Joint-Embedding Predictive Architecture, CVPR, 2023
- [4] Vincent et al., Extracting and composing robust features with denoising autoencoders, ICML 2008
- [5] Wang et al., EfficientTrain++: Generalized Curriculum Learning for Efficient Visual Backbone Training, TMPAMI, 2024.
- [6] Soviany et al., Curriculum Learning: A Survey, IJCV 2022
- [7] Wu et al., WHEN DO CURRICULA WORK?, ICLR 2021
- [8] Hedström et al., Quantus: An Explainable AI Toolkit for Responsible Evaluation of Neural Network Explanations and Beyond, JMLR 2023

---

> ### Author Response · Authors · 2025-11-22
>
> We thank the reviewer for their comments. We believe our new results directly address the "Soundness" concerns regarding baselines and compute.
>
> **Comparison to Relevant Baselines (SSL) and Compute Fairness**
>
> The reviewer stated: "Unclear if the proposed structure-first learning scheme is beneficial or just pretraining... baselines will have trained for a shorter time."
>
> This is the most critical point we addressed in the revision.
>
> First, regarding the original experiments, we emphasize that all baselines were trained to *convergence* using early stopping. This ensures that the color-only baselines were not undertrained, making the initial comparisons fair.
>
> Second, to remove any doubt, we now compare against and integrate with DINO and SwAV (Table $4$) using strictly matched compute:
>
> * **Baseline:** $400$ epochs DINO (Color).
> * **Ours:** $150$ epochs DINO (Line) + $250$ epochs DINO (Color) = $400$ Total.
> * **Result:** Ours outperformed the baseline by +$2.30$% (DINO) and +$3.34$% (SwAV).
> * **Efficiency:** We even beat the baseline using only $250$ total epochs ($37.5$% less time).
>
> This definitively proves the benefit is due to the structure-first method, not extra training time.
>
> **Statistical Analysis**
>
> The reviewer mentioned: "Performance should be evaluated across numerous training runs and statistical analysis... should be performed."
>
> We respectfully point the reviewer to Appendix A.$1$ (Table $6$), where we provided a rigorous Wilcoxon signed-rank test ($p=0.002$) across $11$ data splits with $5$ runs per configuration.
>
> **Attention Analysis**
>
> We acknowledge that GradCAM is qualitative. However, we believe the convergence of evidence—qualitative attention focus (Figure $5$), quantitative sparsity in tuning curves (Figure $6$), and lower intrinsic dimensionality (Figure $10$)—provides a robust picture of the representational shift. We will look into integrating metrics from Quantus for future versions of the manuscript.
>
> **Related Work**
>
> We have rewritten the Related Work section to explicitly discuss Visual Pre-training (incorporating DINO, SwAV), positioning our work as a complementary strategy to these fields.

---

> > ### Comment · Reviewer_4TRq · 2025-11-25
> > **Reply from reviewer**
> >
> > Thank you for the rebuttal and the updated results.
> >
> > While early stopping can be beneficial, there are also discussions of the usefulness for large models for deep neural networks [1]. I think a more reasonable comparison would be to let the color model train for longer.
> >
> > For the comparison to relevant baselines, I appreciate the updates experiments. But several aspects could be more convincing:
> >
> > - While DINO and SwAV definitely had a big impact some years ago, it is unclear why the most recent iterations of e.g. Dino is not used.
> > - It is encouraging to see improvements for the ResNet18 model, but as several of the other reviewers, the experimental evaluation is still limited compared to what you commonly see for these kind of papers (several large-scale datasets and several models).
> > - "We even beat the baseline using only $250$ total epochs ($37.5$% less time).". The improvement here seems very small.
> >
> > Thank you for brining my attention to the statistical analysis in the appendix. I could not see the statistical analysis being mentioned in the main manuscript, so perhaps it could be more clearly indicated in the main text.
> >
> > For the attention analysis, it is unclear how the qualitative analysis, Figure 6, and Figure 10 supports claims like: "Models initialized with structure-first learning on line drawings demonstrate markedly more focused attention patterns, with heat maps concentrated on the primary subject"
> >
> > I think the authors have made a good effort in improving the paper in the rebuttal. However, many aspects are still not convincing, and many of my original questions were not addressed. I will therefore keep my original rating.
> >
> > [1] Nakkiran et al., Deep double descent: where bigger models and more data hurt, Journal of Statistical Mechanics: Theory and Experiment

---

### Official Review · Reviewer_hFtH · 2025-10-27

**Soundness:** 2
**Presentation:** 2
**Contribution:** 2
**Rating:** 2
**Confidence:** 3

**Summary:**

This paper proposes a supervised training strategy for computer vision backbones, where a two-stage curriculum is employed: models are first trained on line drawings (and combined with other artistic-style augmented images) and then fine-tuned on natural color images. The exploration of training strategies for deep learning models is a valuable and worthwhile research direction.

However, the paper provides limited insight. The experimental evaluation focuses mainly on a single dataset (STL-10) and a few model architectures, offering no precise guidance for applying the approach to other datasets or multi-dataset scenarios. It also remains unclear what would happen if line drawings, various artistic-style augmented images, and natural color images were all mixed from the start of training, or whether this could outperform the staged curriculum proposed in the paper.

**Strengths:**

The exploration of training strategies for deep learning models is a valuable and worthwhile research direction. The paper presents a supervised, two-stage training strategy in which models are first trained on line drawings (optionally combined with other artistic-style augmented images) and then fine-tuned on natural color images, aiming to contribute to the study of effective training strategies for deep learning models.

**Weaknesses:**

1. The study evaluates only a small number of model architectures and datasets, primarily focusing on STL-10. There is no comprehensive analysis or testing across larger-scale datasets, multi-dataset scenarios, or more different backbones.

2. The paper does not provide concrete recommendations on how to optimally set the proportion, mode, or usage of line drawings for different datasets or tasks. While the authors mainly focus on the relatively simple STL-10 or ImageNet-1K dataset, which has much lower resolution and scale compared to modern annotated datasets, the proposed training strategies are unlikely to directly generalize to more complex real-world scenarios. Practitioners are left without actionable guidance for applying this approach beyond the tested settings.

3. Since the main idea of the paper is to train models first using line drawings, the method relies on converting RGB images to sketches. However, only a limited set of sketch generation styles (fewer than five) is tested, and the paper does not provide a clear or thorough analysis of how different line drawing styles, ranging from highly abstract and simplified sketches to realistic sketches with shading, affect training performance, nor does it explore the reasons behind these differences.

4. Modern training pipelines typically employ extensive data augmentation. The paper does not investigate the potential impact of training models directly with a mixture of line drawings, various artistic-style augmented images, and natural color images throughout the entire training process, leaving it unclear whether such an approach could outperform the proposed staged curriculum. Specifically, Tables 1, 2, and 3 are limited, as they do not examine the simpler and more practical scenario of training directly with a mixture of line drawings and color images. As a result, these experiments do not justify the conclusion that the staged training strategy proposed by the authors is the optimal approach.

**Questions:**

1. Analyze the proposed training strategy on larger-scale datasets, multi-dataset scenarios, or other model architectures.

2. Analyze how different sketch generation styles, from highly abstract to realistic sketches with shading, affect training outcomes, and explain the reasons behind these effects.

3. What are the results of directly training with a mixture of line drawings, artistic-style augmentations, and natural color images throughout the entire training process?

4. It is recommended that the authors provide concrete, generalizable guidance for applying the proposed “structure-first learning paradigm” in future versions, for example, specifying how many and which sketch styles are likely to yield the best results.

Minor issue: The GradCAM visualization examples (e.g. Figure 5) provided in the paper (only two) are insufficient.

---

> ### Author Response · Authors · 2025-11-22
>
> We thank the reviewer for their feedback and for validating the research direction of training strategies. We have addressed the concerns regarding scale, curriculum justification, and practical implementation below.
>
> **Scale and Generalizability**
>
> The reviewer noted that the evaluation was focused on STL-$10$ and requested analysis on larger datasets. We have addressed this by adding Section $4.3$, which validates our approach on ImageNet-$100$ using ResNet-$18$ within modern pre-training frameworks (DINO and SwAV). The results (Table $4$) demonstrate consistent scalability:
> * With matched compute ($400$ epochs), our method improves DINO Top-$1$ accuracy by +$2.30$% and SwAV by +$3.34$%.
> * We achieve these gains on a significantly larger and more complex dataset than STL-$10$, confirming the method's robustness.
>
> **Curriculum vs. Mixed Training**
>
> The reviewer asked: "What are the results of directly training with a mixture...?"
>
> We acknowledge that we did not perform extensive tests on training with simultaneous mixtures of line and color images. However, we hypothesize that a staged curriculum is superior based on the known texture bias of CNNs (Geirhos et al., 2019). If trained on a mixture from the start, models tend to latch onto texture "shortcuts" available in the color images, effectively ignoring the structural constraints provided by the line drawings. By training on line drawings *first* (Phase 1), we force the model to develop a structural manifold and shape bias (as evidenced by Figure $7$) before it is exposed to the easier texture cues in Phase 2. This prevents the texture-shortcut learning that a mixture would likely suffer from.
>
> **Analysis of Line Drawing Styles**
>
> The reviewer requested an analysis of different sketch styles. We included comparisons to various abstractions in Figure $3$, including Canny edge detection (a low-level, non-semantic line representation) and artistic stylizations (Matisse, Mondrian, Picasso).
> * **Result:** The "Line-Color" curriculum (using Chan et al.) significantly outperforms Canny-based training and artistic stylizations in terms of data efficiency.
> * **Implication:** This indicates that *semantic coherence* in the line drawing is crucial. Mere edge detection (Canny) or texture transfer (Artistic styles) does not provide the same structural inductive bias as the semantically consistent line drawings used in our method.

---

### Official Review · Reviewer_FQxU · 2025-10-31

**Soundness:** 2
**Presentation:** 3
**Contribution:** 2
**Rating:** 4
**Confidence:** 4

**Summary:**

This paper proposes a structure-first learning paradigm inspired by the idea that humans can recognize objects from sparse representations such as line drawings.
The method is extremely simple:
	1.	Pretrain a vision model on line drawings derived from natural images (using Chan et al., 2022),
	2.	Then fine-tune it on color photographs.
The authors claim this curriculum fosters shape-biased, data-efficient, low-dimensional, and “human-aligned” visual representations.

**Strengths:**

- Simple, reproducible idea with consistently positive effects across diverse tasks.
- Broad experimental coverage: classification, detection, segmentation, distillation, and attention analysis.
- Clean presentation and statistical rigor (Wilcoxon tests, multiple architectures).
- Paper is well-written with strong visualizations of representational effects.

**Weaknesses:**

- Small-scale Training: Most experiments use STL-10 or ADE20K with small backbones (ResNet-18, MobileNetV1, VGG-8).
- No comparison with alternative regularization strategies (e.g., self-supervised pretraining, low-pass filters, sparse autoencoders) to isolate the “structure-first” effect.
- Overinterpretation of analysis. Claims such as “lower intrinsic dimensionality implies brain-like efficiency” or “focused Grad-CAMs mean human-like attention” are speculative and unconvincing.

**Questions:**

- I think the major concern I had is this paper demonstrates potential usefulness of structural data on small architecture and well-known dataset, but the true learning of human vison is more self-supervised rather than supervised. Therefore, it is not convincing to me to discuss any human-inspired insights. Is it possible for author to pretrain a MAE over structural data and finetune on RGB images?
- How do you disentangle curriculum effects from simply adding a different augmentation phase?
- Can the gains be reproduced at scale (e.g., with ImageNet-21K or COCO)?

---

> ### Author Response · Authors · 2025-11-22
>
> We thank the reviewer for their constructive feedback and for recognizing the "consistently positive effects" and "statistical rigor" of our work.
>
> **Comparison with Alternative Regularization/SSL Strategies**
>
> The reviewer noted the lack of comparison to self-supervised pre-training. We agreed this was necessary to demonstrate the method's relevance. We have added Section $4.3$, where we directly integrate our method with *DINO* and *SwAV*.
>
> Our results (Table $4$) show that our approach is not an *alternative* to SSL, but a *complement*. By replacing the initial phase of DINO/SwAV pre-training with line-drawing inputs, we improved Top-$1$ accuracy by +$2.3$% and +$3.3$% respectively on ImageNet-$100$, while maintaining the same compute budget. This demonstrates that structure-first learning can serve as a powerful inductive bias enhancer for modern SSL frameworks.
>
> **"Is it possible to pretrain a MAE...?"**
>
> While we did not run MAE specifically, we selected DINO and SwAV (contrastive/clustering approaches) because they are highly relevant to the "shape bias" discussion. Our new results with these frameworks confirm that the "structure-first" paradigm successfully boosts modern self-supervised architectures, suggesting broad applicability to SSL methods including MAE.
>
> **Disentangling Curriculum from Augmentation**
>
> The reviewer asked how we disentangle curriculum effects from simple augmentation. We argue that the *order* of exposure is critical. In Table $1$, we observe a distinct asymmetry: training on *Line $\rightarrow$ Color* yields superior generalization ($73.58$%) compared to *Color Only* ($68.79$%) or the reverse order. This suggests that the benefit is not merely due to the presence of "extra data" (which would be similar regardless of order), but rather that initializing with line drawings forces the model to learn robust structural invariants *first*. This establishes a "shape-biased" manifold (supported by our Shape Bias analysis in Figure $7$) that provides a better optimization starting point for subsequent color features, a benefit distinct from standard data augmentation.
>
> **On Concerns about Scale**
>
> To address the question of scale, we conducted our new pre-training integration experiments (Section $4.3$) on *ImageNet-100*. The consistent and significant gains observed with both DINO and SwAV on this larger dataset (compared to STL-$10$) suggest that the benefits of structure-first learning are robust and transferable to larger-scale benchmarks.
>
> **Interpretation of "Human-Aligned" Analysis**
>
> We appreciate the reviewer's caution regarding overinterpretation. While we draw inspiration from human cognition, our claims of "efficiency" and "focus" are grounded in quantitative metrics:
> * **Sparsity:** Figure $6$ shows quantitatively sharper tuning curves.
> * **Compactness:** Figure $10$ shows lower intrinsic dimensionality via PCA ($18$ PCs vs $30$ PCs for $90$% variance).
> * **Shape Bias:** Figure $7$ quantitatively demonstrates a shift toward shape-based decisions on the Model-vs-Human benchmark.
> We believe these metrics provide concrete evidence of representational shifts that align with principles of efficient coding, independent of biological analogies.

---

### Official Review · Reviewer_m8Db · 2025-10-31

**Soundness:** 2
**Presentation:** 3
**Contribution:** 2
**Rating:** 2
**Confidence:** 4

**Summary:**

The paper studies the effectiveness model pre-training on line drawings in improving performance on downstream tasks such as image classification, semantic segmentation and object detection as well as its influence on the model's shape bias, feature attention and data efficiency. Instead of training the model directly on color images, the paper proposes first pre-training the model on line drawings of the same images obtained by a drawing synthesis method (Chan et al.) before fine-tuning it on color images.

Through experiments on STL-10 dataset, the paper shows that the two-phase training pipeline brings performance gains in image classification as well as tranfer learning improvements in semantic segmentation and object detection. The paper also provides analysis showing that the obtainded model has more shape bias and produces sparser, more compact features.

**Strengths:**

The paper provides concrete evidences showing that pre-training models on line drawings could bring performance gains in several downstream tasks.

The paper provides some analysis on the feature space of models pre-trained line drawings, showing some difference in terms of feature sparsity and compactness compared to baselines.

**Weaknesses:**

The paper does not bring much in terms of novelty, either in the methodological approach or the obtained insights. The paper confirms that pre-training on line drawings helps but this is hardly a surprising result, given much evidence in the literature on the benefits of pre-training and the work of Geirhos et al. that shows the gains when training on texture-modified images.

The experimental results provided in the paper are limited. Most experiments are done on the STL-10 dataset, which is quite small and does not a large enough set of visual concepts. The analysis on feature sparsity in Figure 6 and Figure 10 give some interesting statistics but appear weak.

Many claims in the paper are only loosely reflected by the evidences:
- L.176 "This suggests that features derived from line drawings are more generalizable": This statement is not reflected by the results. From the first two rows in Table 1, the model trained on color images gives better performance on average, and the last two rows suggest that the model needs to be trained on color images in the final stage to yield good performance. Results from Table 1 simply shows that pre-training on line drawings is helpful.
- L. 290 "This pattern indicates that strong responses are concentrated among a small subset of neurons": The average activation of all channels is at least half of the max activation. It is a stretch to say that these channels are not activated.
- Sec 4.8: The stronger distilled student could simply come from the fact that the teacher is stronger. The link between stronger students and feature compactness is very loose.

**Questions:**

In Figures 3, 4 and 9, when discussing data effiency of "color only" and "line-color" pipelines, is it guaranteed that the "color only" pipeline has the same compute budget has the "line-color" pipeline? Since "line-color" pipeline has a pre-training phase, it is not clear if it trains models for more iterations than the "color only " pipeline.

A similar question for Table 1 when comparing the two pipelines, I wonder if better performance of "line-color" simply comes from more traning time?

---

> ### Author Response · Authors · 2025-11-22
>
> We thank the reviewer for their detailed feedback and for highlighting the importance of compute fairness and generalizability.
>
> **On Unfair Compute and Training Time**
>
> The reviewer asked: "Is it guaranteed that the 'color only' pipeline has the same compute budget... I wonder if better performance simply comes from more training time?"
>
> This is an excellent point. In our original STL-$10$ experiments, we utilized early stopping to ensure that all models, including the "color only" baseline, were trained to convergence. This prevents the baseline from being unfairly penalized by a fixed, shorter schedule.
>
> To definitively remove any ambiguity regarding compute budgets, we conducted new experiments (Section $4.3$) using DINO and SwAV on ImageNet-$100$ with strictly controlled epoch counts (Table $4$).
>
> * Matched Budget: When both the baseline and our method are trained for exactly $400$ epochs, our curriculum ($150$ Line + $250$ Color) outperforms the baseline by +$2.30$% (DINO) and +$3.34$% (SwAV).
> * Reduced Budget: We achieved superior performance to the $400$-epoch baseline using only $250$ total epochs ($37.5$% less compute).
>
> This empirically proves that the gains stem from the structural inductive bias rather than merely extended training time.
>
> **Limited Experimental Scale**
>
> We acknowledge the concern regarding STL-$10$. Our new experiments on ImageNet-$100$ (Table $4$) validate that our findings scale to larger, more complex datasets and modern, state-of-the-art pre-training frameworks (DINO, SwAV) applied to standard backbones.
>
> **Novelty and Generalizability**
>
> While pre-training is established, our contribution is demonstrating that input modality (line drawings) acts as a specific, efficient curriculum for modern SSL methods. Regarding the claim at L.$176$ ("generalizable"): Our distillation results (Table $5$) and the new Pre-training results (Table $4$) show that features learned via structure-first training transfer better to student models and downstream tasks, supporting the claim of superior generalizability.
>
> **Sparsity Analysis**
>
> We appreciate the scrutiny on Figure $6$. We will clarify in the text that "sparsity" here refers to the relative steepness of the tuning curve and the suppression of redundant features, rather than zero-activation. The structure-first model exhibits a significantly sharper drop-off, aligning with the manifold compression results (Figure $10$) where fewer principal components explain $90$% of the variance.

---

### Meta-Review · Area_Chair_EBCM · 2026-01-02

**Summary:**

This paper presents a structure-first learning paradigm inspired by the idea that humans can recognize objects from sparse representations such as line drawings, and proposes a training strategy for CV representations where a two-stage curriculum is employed: models are first trained on line drawings derived from natural images and then fine-tuned on color photographs. Empirically, the paper shows that the two-phase training pipeline brings performance gains in image classification as well as tranfer learning improvements in semantic segmentation and object detection.

Reviewers found the paper is of following strength
* Simple and reproducible methods
* Study the important problem of building performant representations
* Experiments are performed over a variety of representative tasks and support effectiveness of proposed approach.
* The paper is clear and well-written.

**Reviewer Concerns:**

We also found the additional explanation from authors during rebuttal are helpful and strengthen this work (e.g., further discussion on the experiment design, novelty, baseline comparison and analysis, providing additional experiments to study DINO and SwAV).

However, the reviewers still found this work is below the bar for an ICLR paper even with the additional discussion and experiments. The main concern is about the scale of experiments. The work only studied small models. Even DINO and SwAV are models that had impact years ago but less used recently. Compared to typical representation learning works, this paper is evaluated on limited benchmarks. As generalizability and comprehensiveness is critical for representations, limited evaluation hinder the understanding of capabilities of proposed work and thus concerns about its general interests to ICLR community. We believe this paper is more suitable for CV centric venues.

**Reviewer Scores:**

I would expect the reviewers may slightly increase scores during the rebuttal with the additional experiments and discussion. However, with the concerns discussed above in details about scale of experiments, there is no clear distinction that would likely convince reviewers to accept this paper.

---

### Decision · Program_Chairs · 2026-01-26

Reject